# MatchMask: Mask-Centric Generative Data Augmentation for Label-Scarce Semantic Segmentation

## Abstract

Current semantic segmentation models are very data-hungry and require massive costly pixel-wise human annotations. Generative data augmentation, which scales the train set using generative models, provides a potential remedy. However, existing text-centric methods struggle to generate complex in-distribution data due to the limitations of text descriptions. In this paper, we propose MatchMask, a novel mask-centric generative data augmentation approach tailored for label-scarce semantic segmentation. It leverages a few labeled semantic masks to generate diverse, realistic, and well-aligned image-mask training pairs for semantic segmentation models. Specifically, to adapt existing text-to-image models for semantic image synthesis, we first propose a Gradient Probe Method to investigate the role of each layer in the diffusion model. On this basis, we introduce a Layer-Timestep Adaptive Adapter (LT-Adapter) comprising layer-adaptive cross-attention fusion and time-adaptive LoRA scaling to enable efficient adaption for the critical layers. Meantime, we design a robust relative filtering principle to suppress incorrectly synthesized regions. Moreover, the proposed approach is extended to MatchMask++ in the semi-supervised setting to take advantage of additional unlabeled data. Experimental results on VOC, COCO and ADE20K demonstrate that MatchMask remarkably enhances the performance of segmentation models, surpassing prior data augmentation techniques in various benchmarks, $e.g$, 67.5%$\rightarrow$74.3% mIoU on VOC. Our code will be made publicly available.

## 1 Introduction

Semantic segmentation aims to assign pixel-level dense semantic labels for an image. It has been extensively investigated and inspired many downstream applications like autonomous driving and medical imaging. Despite the rapid progress of deep neural networks, training a semantic segmentation model usually requires a large number of images with pixel-level annotations. The pixel-wise manual labeling is costly, laborious, and even infeasible, precluding its deployment in some scenes. To avert the labor-intensive procedure, data augmentation Shorten & Khoshgoftaar (2019); Shorten et al. (2021) is an effective manner to expand the diversity of existing data. However, standard **image-centric** transforms (*e.g*, flip, crop) He et al. (2016); Yun et al. (2019); DeVries (2017) offer limited variation, yielding no substantive novel content.

With the advent of generative models Rombach et al. (2022); Goodfellow et al. (2014), Generative Data Augmentation (GDA) has emerged as a promising alternative. It scales the train set by producing synthetic samples with labels based on advanced generative models. Compared to traditional image-centric data augmentation, GDA offers a more extensive variety of data. Recent advancements in text-to-image diffusion models present phenomenal power in generating highly realistic images from textual descriptions. On this basis, DiffuMask Wu et al. (2023b) uses text-guided cross-attention information to localize class-specific regions and finally obtain a pixel-wise mask. Dataset Diffusion Nguyen et al. (2024) carefully designs the text prompts, cross-attention, and self-attention of SD to produce images paired with their corresponding segmentation masks. DatasetDM Wu et al. (2023a) extends text-guided image synthesis to perception data generation by leveraging the rich latent code of the diffusion model and a unified perception decoder. One key characteristic of these state-of-the-art methods is **text-centric**, which uses text as the sole input condition and

Figure 1: Comparisons between image-, text-, mask-centric data augmentation. Mask-centric paradigm could generate more diverse data than image-centric transforms, and more informative, realistic, well-aligned image-mask pairs than text-centric perspective.

suffers from several drawbacks: **1)** The expressiveness of textual descriptions is limited, struggling to precisely convey intricate scenarios or layouts (*e.g*, scenes in ADE20K). **2)** The choice of text prompts is subjective and heuristic (*e.g*, *"a photo of a [class name] [background description]"* used in DiffuMask), and may not align with the distribution of downstream datasets, producing inferior data for model training. **3)** It is challenging to generate images that accurately match our mental imagery via text prompts alone (*e.g*, missing or unexpected objects, object relation errors). Moreover, such methods often produce misaligned image-mask pairs (*e.g*, *table class* in Fig. 1 and more in Fig. 11), resulting in incorrect supervision for model training. Unlike classification tasks typically involving one class per image, semantic segmentation handles more complex scenes with multiple classes and spatial relationships, demanding higher standards for generated data that text-centric approaches struggle to meet.

To this end, we present a novel paradigm: **mask-centric** generative data augmentation for semantic segmentation via only a few densely labeled samples. We can utilize existing masks as conditions to generate numerous images with diverse characteristics. This mask-centric perspective can produce informative and complex images that better align with the distribution of the target data, making it a more effective data augmentation technique than text-centric methods.

Despite the potential of the mask-centric augmentation pipeline, another underestimated issue is the training of mask-to-image synthesis models with only a small number of labeled samples. Existing state-of-the-art semantic image synthesis models rely on abundant densely annotated image-mask data to align semantic layout and generated image, which contradicts the original goal of generative data augmentation. For example, FreestyleNet Xue et al. (2023) fine-tunes the entire U-Net parameters in Diffusion models, which results in heavy overfitting in the few-shot setting (seeing Fig. 2). One potential solution is to use PEFT (Parameter-Efficient Fine-tuning) techniques such as LoRA Hu et al. (2021). However, despite its popularity, the optimal placement of LoRA layers remains undetermined. Most works apply LoRA empirically to specific layers (*e.g*, encoder blocks Zhang & Agrawala (2023)), which is heuristics and overlooks task-specific differences, leading to sub-optimal results for specific tasks. To customize an effective fine-tuning approach for the semantic image synthesis task, we investigate the contribution of each model layer by proposing the Gradient Probing Method, which analyzes layer-specific gradients during training to evaluate their impact. Our study provides a couple of interesting observations: **1)** Only a minority of parameters are highly correlated with spatial control. These critical parameters are dataset-agnostic and consistent across different datasets (shown in Fig. 3). **2)** The same attention layer in different blocks exhibits varying effects. In general, the initial and final few blocks in U-Net play a more prominent role (seeing Fig. 5). It makes sense because of the high feature resolution of these blocks, which is advantageous for spatial control.

Based on these observations, we propose a simple yet effective Layer-Timestep Adaptive Adaptater (LT-Adapter) for efficient semantic image synthesis adaption using only a tiny densely labeled data set. We design a LoRA-style adapter with 0.7M parameters for the founded critical layers while freezing the original models. In addition, to capture the varied impact of different layers and timesteps in diffusion models, we introduce layer-adaptive cross-attention fusion and timestep-adaptive LoRA scaling. We also present a relative filtering startegy to remove inaccurate regions in generated images, preventing negative effects on segmentation model training.

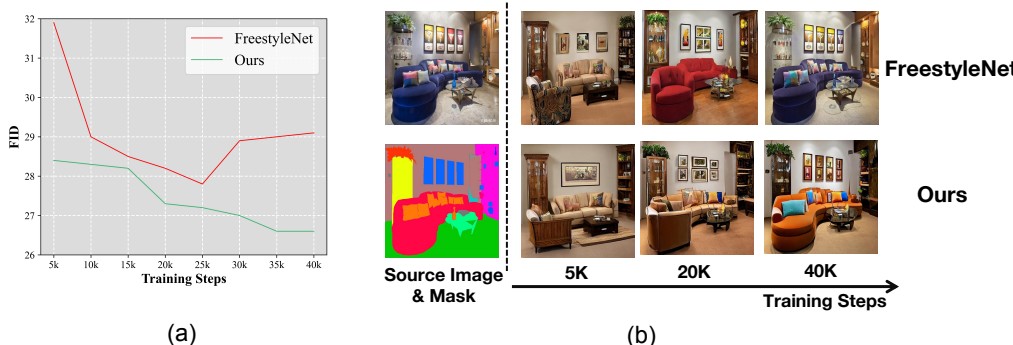

(a)                                          (b)

Figure 2: Quantitative and qualitative results of generated images on ADE20K using 200 samples. **(a):** The trend of FID during training. **(b):** Visualizations of sampled images during training. Existing semantic image synthesis works struggle to work in few sample scenarios, leading to overfitting and loss of pre-trained priors. Our LT-Adapter can learn spatial information well while retain diversity.

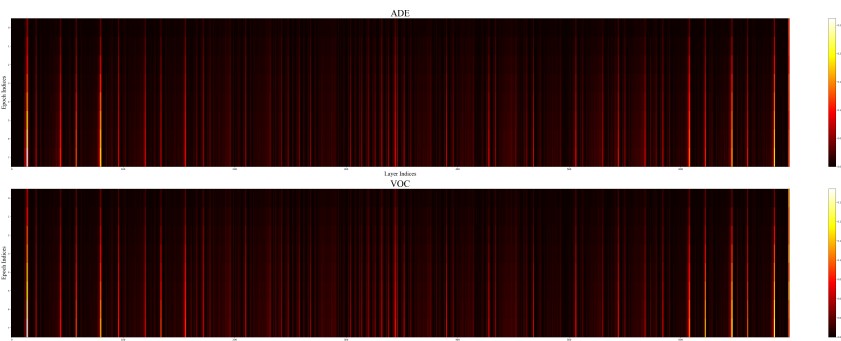

Figure 3: Visualizations of important layers for semantic image synthesis task on ADE and VOC. The x-axis represents the index of each layer in pre-trained models. The y-axis represents checkpoints at different epochs.

Our study mainly focuses on two label-scarce cases: **(1) the data-limited scenario:** with only few labeled image-mask pairs, addressed by MatchMask; **(2) the semi-supervised scenario:** consisting of a limited amount of labeled data in conjunction with additional unlabeled data, solved by MatchMask++. We employ the proposed data augmentation method on a wide range of semantic segmentation benchmarks. It significantly boosts the performance of segmentation models in label-scarce scenarios, achieving over a 4.6% mIoU improvement in the data-limited setting and 6.8% in the semi-supervised setting, surpassing image-centric transform and text-centric methods. Furthermore, MatchMask can be flexibly integrated with existing semi-supervised semantic segmentation methods to enhance their performance (*e.g*, improve Unimatch from 78.3% to 79.6% mIoU on VOC).

Our contributions are summarized as follows:

- **New Roadmap:** We propose MathcMask, a novel mask-centric generative data augmentation for label-scarce semantic segmentation, which can produce more superior training data compared to prior image-centric and text-centric techniques.

- **New Insight:** We observe that only a few fixed layers are critical for the semantic image synthesis task and the varying effect in different blocks by the proposed Gradient Probe Method, which provides valuable insights for future PEFT research.

- **New Method:** We customize an effective fine-tuning approach named LT-Adapter for the semantic image synthesis task, enabling it to work in few-shot settings.

- **Stronger practicality and performance:** Experimental results on various benchmarks demonstrate the effectiveness and superiority of our data augmentation approach.

## 2 RELATED WORKS

### 2.1 GENERATIVE DATA AUGMENTATION

Data augmentation is a universal method to improve the generalization ability of deep neural networks in the case of insufficient training data. Classical data augmentation techniques are performed at the image level, such as geometric transformations He et al. (2016), color space transformations Jurio et al. (2010), and mixing images Hendrycks et al. (2019); Ghiasi et al. (2021). Despite its simplicity, the augmented data lacks diversity due to the reliance on limited source images. With the advance of deep generative models Ho et al. (2020); Rombach et al. (2022), Generative Data Augmentation (GDA) has emerged as a promising alternative. It can produce diverse synthetic samples with labels based on the powerful generative models, which have been widely explored in the classification task Azizi et al. (2023); Kingma et al. (2014); Trabucco et al. (2023); You et al. (2024); He et al. (2022). In the semantic segmentation field, pioneering GAN-based works like DatasetGAN Li et al. (2022a) and BigDatasetGAN Li et al. (2022a) utilize the feature space of pre-trained GANs and design a shallow decoder for generating pixel-level labels for segmentation tasks. However, the quality of the synthesized data is often dissatisfactory due to the limited representation ability of early GAN models. Recent works Nguyen et al. (2024); Wu et al. (2023b) leverage text-to-image models to generate more diverse images paired with masks. However, these works mainly follow the text-centric paradigm, making them challenging to capture complex scenes in semantic segmentation. Additionally, the synthetic data tend to be out-domain and poorly aligned. FreeMask Yang et al. (2024) enhances segmentation via generating mask-based data but is designed for the fully-supervised setting. In this paper, we shed new light on a novel mask-centric GDA paradigm to generate more favorable data for semantic segmentation with only a few densely labeled samples.

### 2.2 LABEL-SCARCE SEMANTIC SEGMENTATION

Semantic segmentation requires very costly pixel-wise human annotations and how to achieve it using scarce annotated samples emerges as a popular topic. Semi-supervised semantic segmentation (SSSS) has been proposed to alleviate the burden of time-consuming manual labeling leveraging limited labeled data along with larger amounts of unlabeled data, including self-training Zou et al. (2020); Yang et al. (2022) and consistency regularization Yang et al. (2023); Wang et al. (2024). Self-training generates pseudo-labels for unlabeled images and leverage them for iterative re-training. Consistency regularization makes predictions invariant to perturbations under different data augmentations. In this paper, we attempt to expand the limited train set by GDA. Apart from SSSS setting, we also consider the more challenging data-limited scene, where no unlabeled data are available.

### 2.3 SEMANTIC IMAGE SYNTHESIS

Semantic image synthesis aims to generate realistic images from given semantic layouts or segmentation maps, including GANs-based Isola et al. (2017); Wang et al. (2018); Park et al. (2019); Liu et al. (2019); Sushko et al. (2020) methods and diffusion-based methods Wang et al. (2022b;a); Nichol et al. (2021); Zhang & Agrawala (2023); Mou et al. (2023); Xue et al. (2023). GAN-based methods tackled this problem under a conditional GAN framework by exploring different conditioning mechanisms in GANs to do stochastic generations that correspond to the input label map. These methods often face challenges in generating diverse and fully controlled images, primarily due to the limitations in the training data and the inherent instability in GAN training processes. Diffusion-based methods mainly rectified specific blocks in the U-Net Ronneberger et al. (2015) architecture to enhance semantic consistency. However, most existing works require a large amount of image-mask pairs for training, while the applicability in scenarios with few training samples are barely explored.

Several works have investigated the utilization of pre-trained models to achieve layout-controllable text-to-image synthesis. Some training-free approaches Parmar et al. (2023); Chefer et al. (2023); Feng et al. (2022); Hertz et al. (2022); Couairon et al. (2023) modify cross-attention maps to control the condition the denoising process but shows inferior performance. Other training-based methods adopt transfer learning to text-to-image diffusion models via either fine-tuning all the parameters Xue et al. (2023); Lv et al. (2024) or introducing and optimizing partial parameters Mou et al. (2023) but heavily rely on abundant annotated data. In this paper, we explore the importance of each layer in pre-trained models and propose a more efficient manner for adaption.

Figure 4: Overview of our framework. The proposed MatchMask includes (1) training LT-Adapter on few labeled samples; (2) mask-to-image data generation; (3) training segmentation model with augmented data. We further present MatchMask++ for the semi-supervised setting to utilize additional unlabeled data.

## 3 METHOD

In this section, we first briefly review the architecture of typical Stable Diffusion Rombach et al. (2022)). Then, we introduce the proposed Gradient Probe Method, LT-Adapter and Relative Filtering Strategy. Finally, we present the pipeline of MatchMask and MatchMask++.

### 3.1 ARCHITECTURE OF DIFFUSION MODELS

The typical Stable Diffusion model consists of an auto-encoder Kingma & Welling (2013) and a conditional latent diffusion model (LDM) Rombach et al. (2022). The auto-encoder enhances efficiency by mapping the pixel space into latent space. The diffusion network is designed in U-Net style, comprising input blocks, middle blocks, output blocks, and other layers. To realize text-to-image generations, the encoded text feature is integrated into the intermediate layers of U-Net via the cross-attention layer. Given text features $\mathbf{c}$ and latent image features $\mathbf{f}$, the cross-attention block first computes $\mathbf{Q} = \mathbf{W}_q\mathbf{f}$, $\mathbf{K} = \mathbf{W}_k\mathbf{c}$, $\mathbf{V} = \mathbf{W}_v\mathbf{c}$ through three projection layers to map them into unified dimension $\mathbf{d}$, and then calculate a weighted sum over value features as:

$$\text{Attention}(\mathbf{Q}, \mathbf{K}, \mathbf{V}) = \text{Softmax}\left(\frac{\mathbf{Q}\mathbf{K}^T}{\sqrt{d}}\right)\mathbf{V} \tag{1}$$

To inject semantics into the layout and achieve semantic image synthesis, FreestyleNet rectifies the attention maps of each text embedding using the corresponding segmentation mask. The processed layout segmentation map $\mathbf{M} \in \mathbb{R}^{H \times W \times C}$, which represents the binary mask for each concept, serves as the attention mask for cross attention to force each concept in the text prompts focus on the corresponding regions. For ease of reading, we denote $\frac{\mathbf{Q}\mathbf{K}^T}{\sqrt{d}}$ as $\mathbf{A} \in \mathbb{R}^{(H \times W) \times C}$, and the rectified cross-attention then turns into:

$$\text{Attention}(\mathbf{Q}, \mathbf{K}, \mathbf{V}) = \text{Softmax}\left(\frac{\hat{\mathbf{A}}}{\sqrt{d}}\right)\mathbf{V}, \text{where} \quad \hat{\mathbf{A}_{i,j}^k} = \begin{cases} \mathbf{A}_{i,j}^k, & \mathbf{M}_{i,j}^k = 1 \\ \text{-inf}, & \mathbf{M}_{i,j}^k = 0 \end{cases} \tag{2}$$

### 3.2 GRADIENT PROBE METHOD

Despite the impressive results of previous semantic image synthesis models, they heavily rely on massive annotated data and struggle to work with limited data due to expensive fine-tuning, *e.g.*, FreestyleNet fine-tunes the entire U-Net model. Research on model pruning Liu et al. (2018); He et al. (2018); Luo et al. (2017) proves that not all parameters are necessary for the downstream task. Here, we propose the Gradient Probe Method to pinpoint the critical parameters for semantic image synthesis in pre-trained diffusion models, which use a few samples to fine-tune Stable Diffusion following FreestyleNet in the early training stage as pre-trained models align layout information before overfitting.

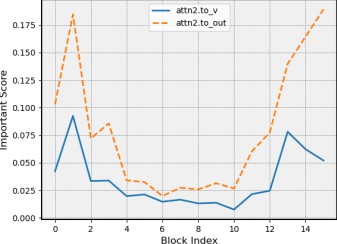
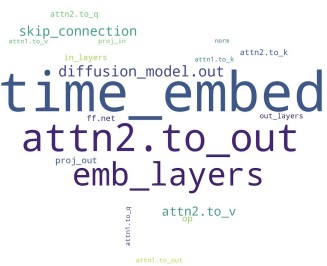

Figure 5: Important scores of *to_k, to_out* in cross attention layer at different U-Net blocks.

Figure 6: Visualizations of important layers.

Gradient descent Amari (1993) is a fundamental optimization algorithm used to minimize a loss function $L(\theta)$ with respect to the model parameters $\theta$. Parameters with consistently high gradient magnitudes are likely to be more influential for the task. The basic update rule for gradient descent is:

$$\theta \leftarrow \theta - \eta \nabla_\theta L(\theta) \tag{3}$$

where $\eta$ is the learning rate, $\nabla_\theta L(\theta)$ is the gradient of the loss function with respect to the parameters.

In the context of a diffusion model, let $\theta = \{\theta_1, \theta_2, \ldots, \theta_L\}$ represent the model parameters, where $\theta_i$ corresponds to the parameters of the $i$-th layer. During training, considering the instability of the gradient at every step, we calculate the epoch-level gradient of each parameter set $\theta_i$ and compute the importance score $S_i$ for each layer:

$$S_i = \frac{\|\theta_i - \theta_i'\|}{\|\theta_i'\|} \tag{4}$$

where $\theta_i'$ denotes the parameters of the $i$-th layer in pre-trained models. Layers with higher scores $S_i$ are considered more critical for the semantic image synthesis task.

We provide the importance score of each layer in Fig. 3, where we can observe that only a few parameters are salient. These salient parameters are consistent across different datasets, which validates that not all parameters are necessary for spatial alignment. The visualization in Fig. 6 indicates that layers related to time and cross attention showcase remarkable priority. Besides, we investigate the cross attention layer (*e.g*, *to_v, to_out*) in different blocks (from the first input block to the last output block). Results in Fig. 5 demonstrate the varying effects of these layers. The initial and final few blocks (with high resolution) in U-Net play a more prominent role than the middle blocks (with low resolution). It makes sense because high-resolution feature maps typically contain sufficient spatial information and are more suitable for rectified cross-attention.

### 3.3 LAYER-TIMESTEP ADAPTIVE ADAPTATER

Based on the identified vital layers, we can perform a more focused model adaptation to enhance efficiency and effectiveness. Rather than directly fine-tuning selected layers, we leverage Low-Rank Adaptation (LoRA) Hu et al. (2021) to adapt these layers dynamically because: 1) LoRA is more parameter-efficient than fine-tuning partial layers (*e.g*, only 0.7M trainable parameters in our setting); 2) LoRA acts as a form of regularization, preventing overfitting and promoting generalization, particularly in few-shot scenarios. Considering the varied impact of different layers and timesteps in diffusion models, we propose a Layer-Timestep Adaptive Adaptater, as shown in Fig. 7.

**Layer-adaptive cross-attention fusion.** Fig. 5 indicate that the same layer parameters in different blocks showcase different impacts on spatial control. Therefore, we introduce a layer-adaptive fusion strategy for cross-attention. Specifically, the input of each cross-attention block is fed into a linear layer to predict an adaptive fusion parameter $\alpha$. Then, the original attention $Attn_{ori}$ (calculated by Eq. (1)) and rectified attention $Attn_{rec}$ (calculated by Eq. (2)) are merged as:

$$\text{Attention} = \alpha \text{Attn}_{ori} + (1 - \alpha) \text{Attn}_{rec} \tag{5}$$

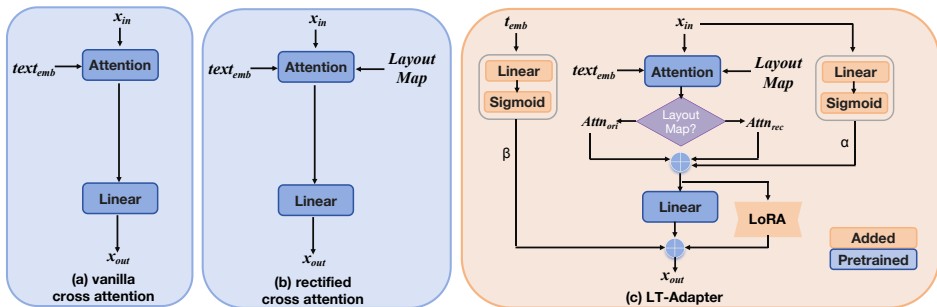

Figure 7: Comparison between the vanilla cross attention, rectified cross attention and our LT-Adapter. We introduce both layer-wise and timestep-wise scale factor for more effective adaption.

This layer-adaptive cross-attention fusion allows each token to interact with both local spatial maps and global contextual information dynamically, facilitating the synthesis of high-quality details.

**Timestep-adaptive LoRA scaling.** Prior work reveals that the reliance on the given condition differs at various timesteps Lv et al. (2024). Motivated by this, we design a timestep adaptive scale factor for LoRA. The time embedding is processed by a linear layer and sigmoid activation to obtain scale factor $\beta$. The LoRA adaption update rule for selected parameters $\theta_{\mathbf{i}} \in \mathbb{R}^{n \times m}$ turns into:

$$\theta'_{\mathbf{i}} = \theta_{\mathbf{i}} + \beta \nabla \theta_{\mathbf{i}} \tag{6}$$

where $\nabla \theta_{\mathbf{i}} = \mathbf{A}\mathbf{B}^{\mathbf{T}}, \mathbf{A} \in \mathbb{R}^{n \times d}, \mathbf{B} \in \mathbb{R}^{d \times m}, d \ll n$ is the LoRA operation.

By incorporating both layer-wise and timestep-wise adaptivity, the proposed strategy enables fine-grained adaptation of critical layers, leading to improved model performance and robustness in few-shot semantic image synthesis tasks.

### 3.4 RELATIVE FILTERING STRATEGY

Despite impressive results, synthetic images still display artifacts in some cases (*e.g*, inferior appearance and unexpected objects in the background in Fig. 8) due to the inherent drawback of pre-trained models, which will degrade segmentation performance during training. A common solution is to filter uncertain pixels by taking the final softmax output as confidence Sohn et al. (2020). However, this approach heavily depends on a trained segmentation model, which is unreliable and accumulates incorrect supervision in data-limited scenarios due to the confirmation bias Arazo et al. (2020).

We present a more robust relative filtering strategy, which identifies the outliers by comparing relative differences across several synthetic images generated by the same mask. The underlying rationale is that these homologous images should yield consistent predictions, regardless of confirmation bias in segmentation models. More formally, given a semantic mask $SM$, we produce $K$ images based on it. Then we predict the pseudo mask $PM_k$ for each image with a segmentation model (trained in label-scarce setting), and perform majority voting to obtain the merged output:

$$\hat{PM}(i,j) = \underset{c \in \{0,1,\ldots,C\}}{\arg\max} \sum_{k=1}^{K} \mathbb{1}\{PM_k(i,j) = c\} \tag{7}$$

where $C$ is the number of classes, $\mathbb{1}$ is the indicator function, $(i,j)$ is pixel index. The merged mask reflects the *prototype* of $K$ images, and we can filter outlier pixels by ignoring the mismatched region between $PM_k$ and $\hat{PM}$. The filtered semantic mask for *k-th* synthetic image turns into:

$$SM_k^{filtered}(i,j) = \begin{cases} SM(i,j), & if PM_k(i,j) = \hat{PM}(i,j), \\ 255, & else \end{cases} \tag{8}$$

By leveraging relative differences across homologous images, noisy regions can be effectively neglected. For example, in Fig. 8 (highlight in the red boxes), the confusing dog paw (in the first row) and redundant sheep (in the second row) are successfully filtered. This voting-style strategy enhances robustness and diminishes the impact of confirmation bias in the segmentation model.

Table 1: Quantitative results of MatchMask on VOC, COCO and ADE dataset in the data-limited setting. The numbers under each dataset represent the fully supervised results.

| Methods | Dataset | Segmenter | Train Data real | synthetic | mIoU | | | |
|---|---|---|---|---|---|---|---|---|
| **Number of Labeled Samples (#):** | | | | | 92 | 183 | 366 | 732 |
| Baseline | | | # | - | 51.7 | 58.6 | 67.5 | 73.6 |
| MatchMask | VOC | DeepLabV3+ | - | 5×# | 52.5 | 58.0 | 66.6 | 70.7 |
| MatchMask | (79.9) | (ResNet101) | # | 5×# | **57.1** | **65.4** | **72.1** | **75.9** |
| Δ ↑ | | | # | 5×# | +5.4 | +6.8 | +4.6 | +2.3 |
| **Number of Labeled Samples (#):** | | | | | 232 | 463 | 925 | 1849 |
| Baseline | | | # | - | 22.3 | 29 | 36.0 | 42.8 |
| MatchMask | COCO | DeepLabV3+ | - | 5×# | 21.2 | 28.3 | 35.8 | 39.7 |
| MatchMask | (57.3) | (ResNet101) | # | 5×# | **24.5** | **32.5** | **39.7** | **44.8** |
| Δ ↑ | | | # | 5×# | +2.2 | +3.5 | +3.7 | +2.0 |
| **Number of Labeled Samples (#):** | | | | | 200 | | | |
| Baseline | | | # | - | 18.6 | | | |
| MatchMask | ADE | Mask2former | - | 5×# | 19.6 | | | |
| MatchMask | (52.4) | (Swin-B) | # | 5×# | **21.4** | | | |
| Δ ↑ | | | # | 5×# | +2.8 | | | |

## 3.5 MATCHMASK AND MATCHMASK++

By leveraging the proposed LT-Adapter and Relative Filtering Strategy, we can generate diverse and high-quality training data using only scarce labeled samples. This data augmentation pipeline in the data-limited scenario is called **MatchMask**. For the relevant semi-supervised setting, which has access to additional unlabeled data, we further introduce **MatchMask++**, as shown in Fig. 4. Specifically, based on a model trained on labeled data (enhanced with MatchMask augmentation), pseudo-labeling is employed to extract more realistic masks from the unlabeled data. These pseudo masks are then used for data augmentation with our adapted mask-to-image model. Notably, 1) there is no need to retrain the LT-Adapter; 2) despite possible noise in the pseudo masks, the augmented image-mask pairs retain excellent quality, as the mask-to-image model aligns its output with the noisy masks (seeing Fig. 9), ensuring effective supervision for the segmentation model.

## 4 EXPERIMENTS

### 4.1 EXPERIMENTAL SETUP

**Datasets and Evaluation Metric.** We conduct experiments on both object-centric PASCAL VOC Everingham et al. (2010), MS COCO Lin et al. (2014) and scene-centric ADE20K Zhou et al. (2017) benchmarks. PASCAL VOC 2012 contains 21 categories (one background category), with 10,582 training images and 1449 validation images. COCO 2017 includes 118,000 training images and 5,000 validation images, covering 80 object categories and a background category. ADE20K includes 150 semantic categories, with 20,210 training images and 2,000 validation images, which are highly challenging due to the complex taxonomy. In few shot settings, we adopt the number of few labeled samples following semi-supervised segmentationYang et al. (2023) for VOC (92/183/366/732) and COCO (232/463/925/1849), and randomly select 200 samples (about 1%) for ADE20K. The mean Intersection over Union (mIoU) is adopted as the evaluation metric in our experiments.

**Implementation Details.** For mask-to-image synthesis, we adopt the Stable Diffusion V1-4 Rombach et al. (2022) as the pre-trained model, and only fine-tune the proposed LT-Adapter with a batch size of 4. We train for 100k iterations with a base learning rate of 4e-5. Image generation uses 50 PLMS Liu et al. (2022) sampling steps with a classifier-free guidance Ho & Salimans (2022) scale of 2. These experiments can be completed on a single NVIDIA A100 GPU with 80GB memory. For the semantic segmentation task, all our experiments were conducted within the mmsegmentation Contributors (2020) framework to ensure consistency and fairness in comparison. We adopt DeepLabv3+ Chen et al. (2018) model with ResNe101 He et al. (2016) for VOC and COCO, and Mask2former Cheng et al. (2022) with Swin-B Liu et al. (2021) for ADE20K dataset. The synthetic image number for each mask is set to 5 (K=5) as default. More details are presented in the Appendix.

Table 2: Comparison to text-centric methods on VOC with 366 labeled samples.

| Methods | Paradigm | Synthetic Data | Synthetic Only | Joint Training |
|---|---|---|---|---|
| Dataset-Diffusion | | 40k | 54.2 | 62.5 |
| Dataset-Diffusion* | Text-centric | 40k | 58.5 | 64.3 |
| DatasetDM | | 40k | 60.2 | 68.2 |
| MatchMask | Mask-centric | 1.1k | 64.6 | 71.4 |
| MatchMask | Mask-centric | 1.8k | **66.6** | **72.1** |

## 4.2 MAIN RESULTS

**Data-Limited Setting.** We first report the results of MatchMask in the data-limited setting, where only a few labeled samples are available. The baseline model is trained only using few densely annotated real training images, and then compared to the counterpart of using synthetic images alone. As shown in Tab. 1, on all three datasets, using synthetic training images yields comparable or even better results than real training images (*e.g*, 51.7% vs. 52.5% on VOC). The performance can be further boosted by joint training using both real and synthetic images. For example, the maximum improvement for VOC, COCO, and ADE is 6.8%, 3.7%, and 2.8% mIoU, validating the superiority of this data augmentation method.

Table 3: Comparison to image-centric methods on VOC with 366 labeled samples. $\mathcal{R}$ and $\mathcal{S}$ represent real and synthetic data, respectively.

| Methods | Train Data | mIoU |
|---|---|---|
| crop | | 64.8 |
| crop,flip | $\mathcal{R}$ : 366 | 65.9 |
| crop,flip,color | | 67.5 |
| crop,MatchMask | $\mathcal{R}$ : 366 | **71.5** |
| crop,flip,MatchMask | + | **72.0** |
| crop,flip,color,MatchMask | $\mathcal{S}$ : 1.8$k$ | **72.1** |

Table 4: Results of MatchMask in the semi-supervised setting.

| Method | VOC | ADE |
|---|---|---|
| *Without unlabeled data* | | |
| Baseline | 67.5 | 18.6 |
| +MatchMask | 72.1 | 21.4 |
| *With unlabeled data* | | |
| Self-Training | 72.6 | 22.7 |
| +MatchMask | 73.5 | 22.9 |
| +MatchMask++ | **74.3** | **24.6** |

**Comparison with other data augmentation methods.** Tab. 2 presents a comparison between MatchMask and text-centric augmentation techniques. Dataset-Diffusion and DatasetDM rely on text prompts generated by GPT-4 Achiam et al. (2023), while Dataset-Diffusion* employs an image captioning model BLIP Li et al. (2022b) to generate captions for all images in the dataset. Despite using a much larger synthetic dataset (40k images), the text-based methods show a substantial performance gap compared to MatchMask, which leverages only 1k-2k images. The superior performance of MatchMask can be explained by 1) its ability to produce more informative and realistic images and 2) the higher accuracy of image-mask pairs created in a mask-centric manner.

In Tab. 3, we compare MatchMask with traditional image-centric augmentation techniques, such as cropping, flipping, and color jitter following Wu et al. (2023a). The results suggest that image-level transformations provide only marginal improvements due to the constrained variability of the source images. However, when integrated with MatchMask, performance increases significantly by 6-7% in mIoU, highlighting MatchMask's superior ability to to enhance data diversity.

**Semi-supervised setting.** In this setup, we propose MatchMask++ to benefit from additional unlabeled data. Our baseline follows a self-training strategy, using labeled data along with pseudo masks on unlabeled data, which are predicted by a model trained on the labeled data (MatchMask augmented). Results in Tab. 4 indicate that MatchMask excels in both cases, and MatchMask++ can further enhance the results to 74.3% and 24.6% IoU on VOC and ADE using unlabeled data.

Table 5: MatchMask combined with Unimatch.

| Method | Unimatch | +MatchMask | +MatchMask++ |
|---|---|---|---|
| mIoU | 78.3 | 79.0 | **79.6** |

Table 6: Ablation study of filtering strategies.

| Strategy | Original | Confidence Filter | Relative Filter |
|---|---|---|---|
| mIoU | 64.7 | 64.9 | **66.6** |

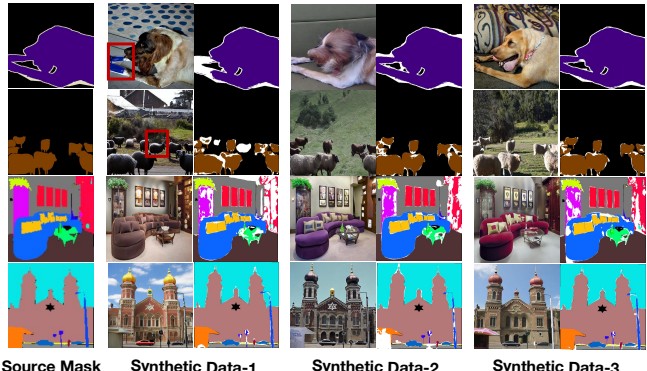

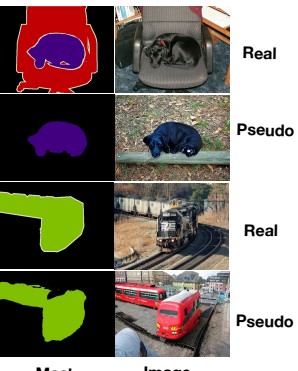

Figure 8: Visualizations of augmented data pairs generated by source masks with noise filtering on VOC and ADE.

Figure 9: Visualizations of real/pseudo mask based generation.

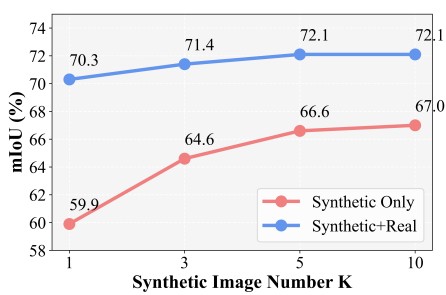

Figure 10: Ablation study on the hyper-parameter K.

| Adaptive Strategies | | Train | Val |
|---|---|---|---|
| Layer | Timestep | | |
| | | 57.2 | 47.2 |
| ✓ | | 58.0 | 47.5 |
| | ✓ | 58.3 | 47.8 |
| ✓ | ✓ | **58.7** | **48.0** |

Table 7: Ablation study of adaptive finetuning strategies proposed in LT-Adapter on ADE.

**Intergration with SSSS methods.** As a data augmentation method, MatchMask has the potential to integrate with existing semi-supervised semantic segmentation (SSSS) works to achieve better results. In Tab. 5, we integrate MatchMask(++) with the classical Unimatch on VOC utilizing 366 labeled samples. The results reveal a substantial improvement in SSSS performance, which is comparable to fully supervised(79.6% vs. 79.9% mIoU).

### 4.3 ABLATION STUDIES

**Effect of relative filtering strategy.** In Tab. 6, we analyze the impact of relative filtering strategy and compare it to confidence-based filtering. The results show that it delivers a notable improvement in performance, outperforming confidence-based techniques. It proves that our majority voting-based method is more robust and suffer less from the confirmation bias in unreliable models.

**Effect of adaptive adaption strategies.** In Tab. 7, we analyze the effect of the proposed layer-adaptive and timestep-adaptive strategies on ADE. Inspired by Tumanyan et al. (2022), we evaluated image quality by calculating the similarity between real and generated images using DINO's dense token representations. The results reveal that both strategies independently enhance performance, with their combination leading to better results. We compare more fine-tuing strategies in Appendix.

**Effect of the number of synthetic images.** As present in Fig. 10, with the increase in synthetic data, the model performance incrementally improves but eventually plateaus. To balance the trade-off between performance and efficiency, we default to generating five images for each mask in this work.

## 5 CONCLUSION

In this paper, we present MatchMask, a novel mask-centric generative data augmentation approach for label-scarce semantic segmentation. By leveraging the proposed Gradient Probe Method, LT-Adapter and Relative Filtering Strategy, we can scale the train set with diverse and high-quality data. It significantly enhances the segmentation performance in both data-limited and semi-supervised scenarios. We hope it would offer novel insights for the community.

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

## A    MORE IMPLEMENTATION DETAILS

**Semantic image synthesis:** During training, we resize all semantic masks to 512x512 following FreestyleNet Xue et al. (2023). We freeze the entire model and only fine-tune the proposed LT-Adapter with LoRA rank set to 4. The training can be completed within 27 hours for 100k iterations on a single A100 GPU. In the mask-to-image synthesis stage, the output images are resized to the original resolution of the input masks, with each mask generating 5 images as default.

**Semantic segmentation model training.** All the semantic segmentation models are trained in the mmsegmentation codebase. For VOC and COCO, we adopt DeepLabv3+ Chen et al. (2018) model with ResNe101 He et al. (2016), and Mask2former Cheng et al. (2022) with Swin-B Liu et al. (2021) for ADE20K dataset. Images are randomly scaled to [0.5, 2.0] and cropped to 512x512. The batch size is set to 8 for each GPU, and a total of 2 GPUs are used. We use AdamW optimizer with 1e-4 weight decay. The initial learning rate is 1e-4, with the polynomial learning rate decay $lr_{iter} = lr_{init}(1 - \frac{iter}{maxiter})^\gamma$, where $\gamma = 0.9$. In the data-limited setting, we train VOC (92/182/366/732) for 3k/5k/8k/10k iterations, train COCO for 20k iterations and train ADE for 10k iterations. For the semi-supervised setting, we extend training iterations to 20k for VOC and ADE. We use the default image augmentation in mmsegmentation, such as RandomCrop, RandomFlip and PhotoMetricDistortion.

Table 8: Comparison to advanced image transforms on VOC.

| Methods | baseline | Cutout | Mosaic | MatchMask |
|---------|----------|--------|--------|-----------|
| mIoU | 67.5 | 68.1 | 67.6 | **72.1** |

## B    ADDITIONAL EXPERIMENTS

### B.1    COMPARISON WITH ADVANCED IMAGE TRANSFORMS

In this section, we compare MatchMask with stronger image-level data augmentation, including Cutout DeVries (2017) and Mosaic Bochkovskiy et al. (2020). Cutout randomly drops some regions of the image. Mosaic combines the four images given into one output image. The output image is composed of the parts from each sub-image. We compare them to our MatchMask on VOC with 366 labeled data. Results in Tab. 8 indicate that the performance gains from these two methods are minimal and significantly inferior to our solution, validating the superior ability of MatchMask in the label-scarce scenario.

### B.2    COMPARISON WITH OTHER FINE-TUNING STRATEGIES

**Baseline.** Since our LT-Adapter is the extension of FreestyleNet in the few-shot setting, we treat it as the baseline, which fine-tunes the entire U-Net. In addition, we incorporate three common fine-tuning methods: (1) fine-tuning residual blocks in U-Net; (2) fine-tuning transformer blocks in U-Net; (3) fine-tuning critical layers selected by our Gradient Probe Method. To be consistent with FreestyleNet, we conduct experiments on ADE20K, randomly selecting 200 training samples (about 1%) as labeled data and evaluating on the validation set to verify whether the model is overfitting.

**Evaluation Metrics.** To ensure comparability with FreestyleNet, we adopt Fréchet Inception Distance (FID) Heusel et al. (2017) to assess the visual quality of generated images. To measure layout consistency, unlike previous methods using a pre-trained segmentation model (*e.g*, UperNet101 Xiao et al. (2018)) on the specific dataset to calculate mIoU between GT and predictions, which suffers from model's low performance and limited generalization, we propose a more effective evaluation metric termed DINO-SIM. It is based on DINOv2 Oquab et al. (2023), which is a large pre-trained self-supervised model and has powerful representation capabilities. The pixel-level features learned by DINOv2 are well-suited for dense prediction tasks such as semantic segmentation. We use DINOv2 to extract dense features from the generated/original images (resized to high resolution $840 \times 840$) and compute the cosine similarity between these features to measure layout consistency. This method provides improved accuracy and generalization compared to traditional metrics.

**Results.** In Tab. 9, we can observe that 1) Transformer blocks play a more significant role in spatial information control than residual blocks, even though they have fewer trainable parameters; 2) The

Table 9: Quantitative results on the ADE20K in the few samples setting.

| Methods | Trainable Params. | FID ↓ | DINO-SIM ↑ |
|---|---|---|---|
| *Few Samples (1%):* | | | |
| UNet | 859M | 29.1 | 44.7 |
| Residual block | 615M | 32.2 | 40.1 |
| Transformer block | 242M | 27.9 | 45.6 |
| Critical layer | 72M | **27.4** | 46.2 |
| LT-Adapter (Ours) | 0.7M | 27.6 | **48.0** |
| *All Samples:* | | | |
| UNet | 859M | **26.2** | 49.4 |
| LT-Adapter (Ours) | 0.7M | 27.1 | **49.4** |

identified critical layers prove to be effective, yielding better results than incorporating task-irrelevant layers; 3) Our LT-Adapter attains better layout consistency (higher DINO-SIM) with a remarkably small number of parameters; 4) When using all samples, our model's performance continues to improve, demonstrating a scalable capability. With only 1% of the training data and 0.7 million trainable parameters, our method achieves performance comparable to fully-supervised methods.

## C  DISCUSSIONS AND LIMITATIONS

**Discussions:** In this work, we propose a novel mask-centric generative data augmentation paradigm for label-scarce semantic segmentation. We have validated its superior performance on various benchmarks. The key design is the LT-Adapter to achieve mask-to-image synthesis via only few densely labeled samples, which is based on the pre-trained Stable Diffusion model. We believe the semantic image synthesis process can be further enhanced by adopting more powerful generative models in the future. In addition, we directly produce $K$ images for each mask in our experiments while not focusing on the value of each sample for the model training. A potential improvement direction is to select more hard samples for segmentation models (*e.g*, the classes showcase inferior performance).

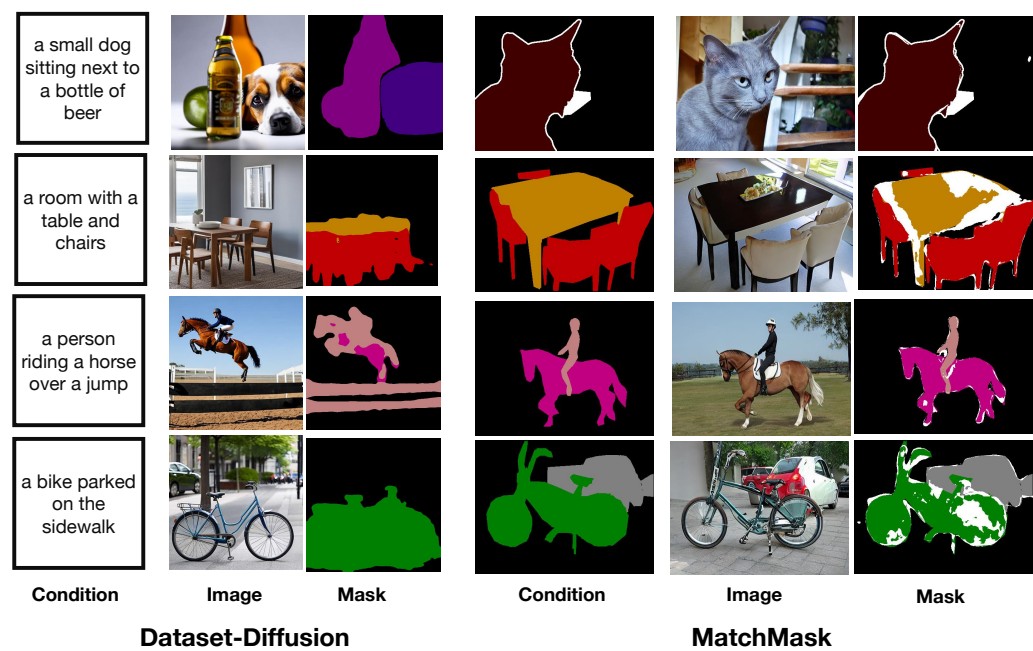

Figure 11: Visualizations of text-centric and mask-centric generative data augmentation.

**Limitations:** While MatchMask has shown impressive performance for data augmentation, it still has some limitations. First, the inference speed during image synthesis is unsatisfactory (*e.g*, 4s/image on VOC or 6s/image on ADE) using PLMS sampling for 50 steps. This stems from the inherent drawbacks of the diffusion model, and we believe future advances in generative paradigms or samplers could address this. Additionally, although MatchMask could produce images based on the pseudo mask, the synthetic image would become messy and unrecognizable if the mask is too noisy. The potential solutions include combining with semi-supervised works to obtain more accurate pseudo masks or designing some principles to neglect these unreliable samples.

## D    MORE VISUALIZATIONS

In Fig. 11, we provide additional qualitative results generated by our mask-centric MatchMask and text-centric Dataset-Diffusion Nguyen et al. (2024). We can observe that 1) Dataset-Diffusion sometimes generates delusive images that are not in the same domain as labeled images. This is the intrinsic limitation of text-centric techniques because it is not trivial to ensure it via text alone. In contrast, MatchMask fine-tunes LA-Adapter on a few source images, yielding in-domain images. 2) For the synthetic image-text pairs, MatchMask showcases higher quality than Dataset-Diffusion. This is because Dataset-Diffusion follows a *generate image then annotate it* manner, which leverages cross-attention between the words in the text and regions in the generated image to produce mask annotation, yielding inferior performance. Differently, MatchMask follows the *generate image to match given mask* manner, leading to more reliable training data.

In Fig. 12, we present the multiple images generated by the same mask on VOC, COCO and ADE. Results show that MatchMask can produce images with diverse contents and styles, which is beneficial for data augmentation.

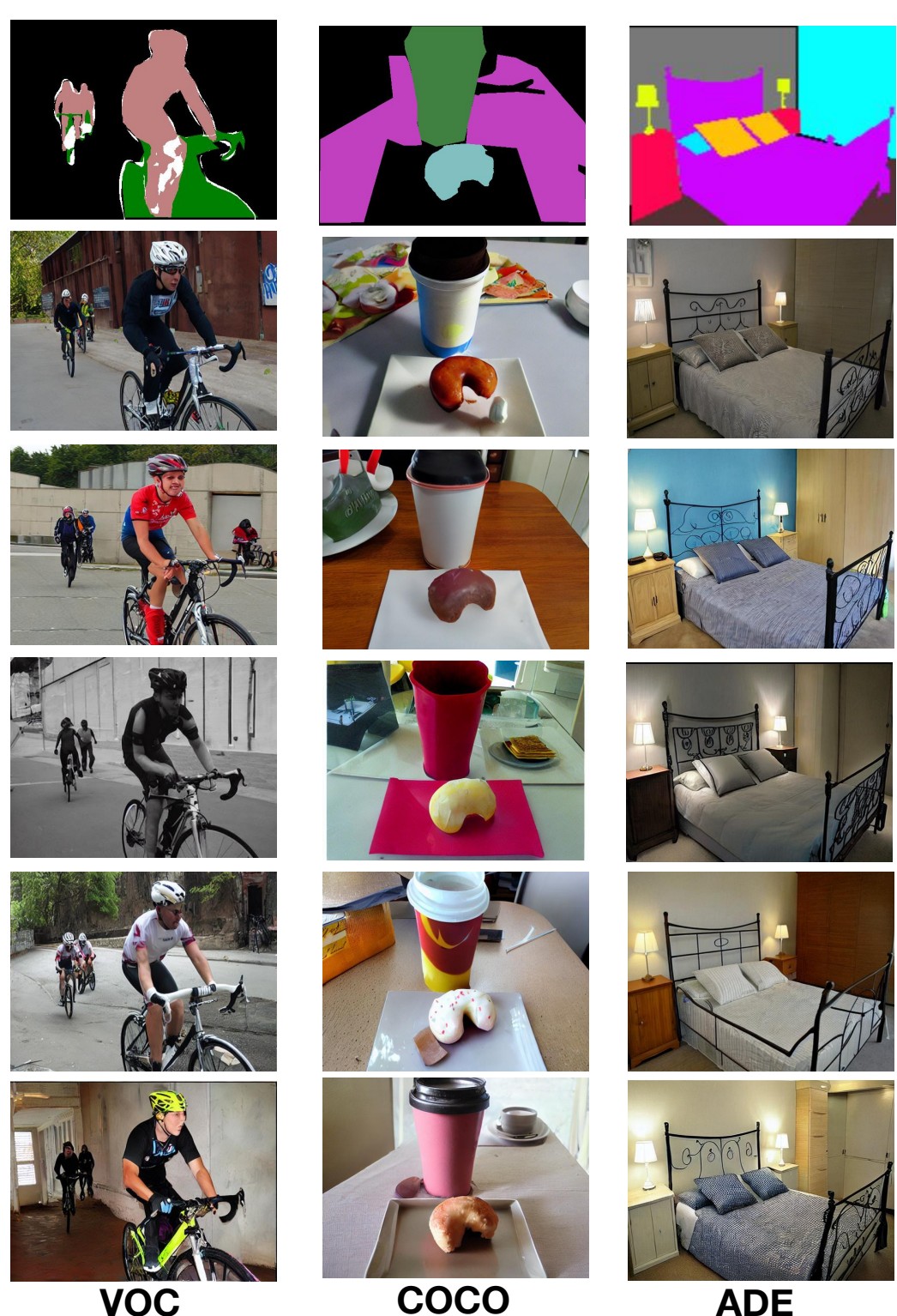

**VOC**  **COCO**  **ADE**

Figure 12: Visualizations of diverse synthetic images on VOC, COCO and ADE.

