# OpenReview forum: "MatchMask: Mask-Centric Generative Data Augmentation for Label-Scarce Semantic Segmentation"
_ICLR.cc/2025/Conference — ICLR 2025 Conference Withdrawn Submission_

### Official Review · Reviewer_DzHi · 2024-10-27

**Soundness:** 3
**Presentation:** 3
**Contribution:** 1
**Rating:** 3
**Confidence:** 4

**Summary:**

This paper presents MatchMask, a novel mask-centric generative data augmentation framework designed for semantic segmentation tasks in label-scarce scenarios. Traditional semantic segmentation models often rely heavily on large datasets with dense pixel-level annotations, which are labor-intensive to obtain. To address this challenge, the authors propose a method that leverages a limited number of labeled masks to generate diverse and well-aligned image-mask pairs, providing effective training data for segmentation models.

**Strengths:**

This paper demonstrates strengths in writing quality, with a clear organization and concise expression that aid in understanding the proposed methodology and experimental results.

**Weaknesses:**

1. Lack of Novelty: This paper applies existing diffusion models to semantic segmentation tasks in label-scarce scenarios, which has already been explored in related works [1-4]. More importantly, compared to these works, the approach proposed here does not exhibit significant technical innovation, particularly in the Layer-Timestep Adaptive Adapter (LT-Adapter) module. This module lacks a clear uniqueness and advantage over existing methods, especially in the context of fine-tuning diffusion models with pseudo-labels, e.g., [3].
2. Unclear contribution: The paper does not fully clarify the specific technical advances that make this work better than [3] and [4]. The improvement of this work may simply come from the additional sampling of the diffusion model on pseudo-labeled images rather than from any innovative technical improvements, and it seems that the [3][4] techniques can also be used to sample on pseudo-labeled data. From this perspective, the comparison in Table 2 does not seem to be entirely fair, because neither [1] nor [2] use pseudo-labeled data for sampling, and even [2] does not fine-tune the diffusion model at all. In addition, there is no clear relationship between LT-Adapter and fine-tuning with pseudo-labels, which makes the contribution of this work seem ambiguous and less influential.
3. Unclear technical details. Moreover, it is difficult to see the improvement in the principle of LT-Adapter's fine-tuning method compared to [3]'s method, and what definite benefits it brings.  Pseudo-labels are full of semantic noise. How to ensure semantic inconsistency when using pseudo-labels for sampling?

Overall, this paper seems to be an application of some technologies, with weak innovation, lack of substantial technical advantages and major contributions and fails to convincingly demonstrate the practical value and necessity of this method compared with existing methods.

[1] DatasetDM: Synthesizing Data with Perception Annotations Using Diffusion Models

[2] Dataset Diffusion: Diffusion-based Synthetic Dataset Generation for Pixel-Level Semantic Segmentation

[3]  Freestyle layout-to-image synthesis.

[4] Freemask

**Questions:**

Please see the Weaknesses

---

### Official Review · Reviewer_hihT · 2024-10-31

**Soundness:** 1
**Presentation:** 2
**Contribution:** 2
**Rating:** 3
**Confidence:** 5

**Summary:**

This paper proposes a pipeline that adapts StableDiffusion for generative data augmentation in few-shot semantic segmentation.

**Strengths:**

1. The experiments show that MatchMask and MatchMask++ consistently outperform the baselines on few-shot and semi-supervised semantic segmentation.
2. MatchMask++ shows that this approach can benefit from more unlabeled data.
3. The fine-tuning is parameter-efficient with its proposed variant of LoRA for StableDiffusion.

**Weaknesses:**

1. This paper over-claims its contributions. The idea of adapting generative model to few-shot semantic segmentation is not novel. In my knowledge, DDPM-Seg [1] is the much earlier work that adapts diffusion generative model on few-shot settings. DatasetDDPM is a baseline that is similar to DatasetGAN included in [1]. This work should cite [1] and compare to DDPM-Seg and DatasetDDPM.
2. The presentation is poor. For the proposed Gradient Probe Method, this paper hardly explains why it is valid or where it comes from. About the significance of layers, many work simply illustrate by visualizing PCA of attention maps or features at different layers. For example, [1] and many recent work that cite [1] reach the consensus that the early layers in SD UNet capture semantic layout, while the later layers in SD UNet capture details. Moreover, the word cloud of Fig.6 does not help understanding layers in any sense.
3. This work builds upon a generative foundation model, StableDiffusion, to reach improvements on datasets like VOC, COCO, and ADE20k. Such datasets have small domain gaps to the source of StableDiffusion. However, it is unclear if this approach can be generalized to other domains, such as medical images.
4. This pipeline seems transferrable to fully-supervised setting. What are the performances in fully-supervised setting, comparing to FreeMask under the same task? This is important since the current problem of definition of "label-scarce" is vague.

[1] Label-Efficient Semantic Segmentation with Diffusion Models. ICLR 2022

**Questions:**

The followings seem to be typos, which do not affect my ratings:
1. line 152, "We propose MathcMask" should be "We propose MatchMask"
2. line 107, "relative filtering startegy" should be "relative filtering strategy"
3. line 513, "Intergration" should be "integration"
4. line 476, "to to" is a grammar mistake
5. Though "adaption" is considered correct by dictionary, "adaptation" is typically used in academic writing.

---

### Official Review · Reviewer_6kUZ · 2024-11-03

**Soundness:** 3
**Presentation:** 3
**Contribution:** 2
**Rating:** 5
**Confidence:** 3

**Summary:**

This paper proposes using a mask-centric data generation approach to address the issue of insufficient data for rare categories, replacing traditional text-centric data generation methods. Extensive experiments demonstrate the effectiveness of the proposed data augmentation pipeline.

**Strengths:**

1. The article is clearly written and easy to understand.

2. The motivation is clear: mask-centric generative data augmentation can produce samples with more accurate semantic masks compared to text-centric methods.

3. Extensive experiments demonstrate the effectiveness of the proposed data augmentation pipeline.

4. Moreover, the proposed method shows improved performance compared to several state-of-the-art methods across multiple benchmarks.

**Weaknesses:**

1. Compared to text-centric methods, MatchMask relies on ground truth masks or pseudo masks to generate data, which can be difficult to collect in real applications. This implies that MatchMask may struggle in scenarios requiring the generation of larger quantities of data.

2. Although MatchMask achieves commendable results, its innovation is somewhat limited; it resembles a mask-conditioned ControlNet combined with some parameter-efficient fine-tuning techniques.

3. The categories in VOC, COCO, and ADE20K are generally quite common and fall within the training distribution of the pre-trained diffusion model. However, when faced with more realistic low-data scenarios, such as medical images or rare categories in remote sensing images, it remains uncertain whether MatchMask can still perform effectively with minimal parameter fine-tuning.

**Questions:**

See weaknesses

---

### Official Review · Reviewer_VjVA · 2024-11-03

**Soundness:** 2
**Presentation:** 3
**Contribution:** 1
**Rating:** 3
**Confidence:** 5

**Summary:**

This paper introduces a method called MatchMask that includes training a semantic image synthesis model (conditioned on segmentation masks), generating synthetic images based on GT masks, training a segmentor on combined real and synthetic data,  and optionally using the trained segmentor to pseudo label unlabeled data. The author demonstrates promising results of MatchMask on both limited data and semi-supervised settings.

**Strengths:**

- Paper is well written, and easy to understand. Sufficient details are provided.
- There’s two variants of MatchMask to serve different needs, making the proposed method more generally useful.

**Weaknesses:**

- The paper mentions FreeStyleNet as an existing method to do semantic image synthesis but there’s no experimental  comparison with FreeStyleNet. ControlNet and other T2I adapters are also good choices when it comes to controlled image generation but the paper fails to acknowledge and compare with those methods. Designing and applying adapters to text2image diffusion models is nothing new. If the intention is to introduce a new adapter as a contribution, there should be a more comprehensive evaluation and comparison against existing adapters.

- Besides the layer/time-wise adapters which are seemingly new to the community (or they may be not), the rest of the paper is mostly about fairly straightforward and obvious findings. For example, generating pseudo labels on unlabeled data is a well tested approach that everyone expects it to work well. One of the paper’s claimed contributions in Sec1 is about providing new insight on which layers should be fine tuned but this is rather unrelated to the semantic image synthesis problem/task at hand (it’s more about PEFT in general). There’s nothing about the adapter design that is motivated or inspired by the actual problem.

 - The benefits of MatchMask are only demonstrated in the limited data settings and on semantic segmentation. There’s other segmentation tasks like instance and panoptic segmentation which may not be helped by MatchMask. In addition, many real-world applications allow segmentation models to be trained on some reasonable amount of real data. It would be stronger if the paper could show that segmentation models to leverage the entire  ADE/COCO real train dataset while getting further improvement with additional synthetic data. Currently, the settings are very weak and somehow unrealistic.

- There’s limited or no comparison with existing works that employ text2image diffusion models to generate data (mask -> image) for segmentation models. This line of work is not new and there have been quite a number of existing works including DiffuMask, FreeMask, SegGen.


- Typo in Sec 3.3 header. “Adaptater”

**Questions:**

How do you balance the number of training samples between real and synthetic?

---

### Note · Authors · 2024-11-15

I have read and agree with the venue's withdrawal policy on behalf of myself and my co-authors.